# Resveratrol Pretreatment Ameliorates Concanavalin A-Induced Advanced Renal Glomerulosclerosis in Aged Mice through Upregulation of Sirtuin 1-Mediated Klotho Expression

**DOI:** 10.3390/ijms21186766

**Published:** 2020-09-15

**Authors:** Chin-Chang Chen, Zi-Yu Chang, Fuu-Jen Tsai, Shih-Yin Chen

**Affiliations:** 1Department of Traditional Chinese Medicine, Chang Gung Memorial Hospital, Keelung 204, Taiwan; geoge6211@gmail.com (C.-C.C.); changzhi887@gmail.com (Z.-Y.C.); 2Department of Anatomy, School of Medicine, China Medical University, Taichung 404, Taiwan; 3Institute of Traditional Medicine, School of Medicine, National Yang-Ming University, Taipei 112, Taiwan; 4School of Chinese Medicine, China Medical University, Taichung 404, Taiwan; d0704@mail.cmuh.org.tw; 5Genetics Center, Medical Research, China Medical University Hospital, Taichung 404, Taiwan; 6Department of Medical Genetics, China Medical University Hospital, Taichung 404, Taiwan

**Keywords:** resveratrol, sirtuin 1, klotho, glomerulosclerosis

## Abstract

Aging kidneys are characterized by an increased vulnerability to glomerulosclerosis and a measurable decline in renal function. Evidence suggests that renal and systemic klotho and sirtuin 1 (SIRT1) deficiencies worsen kidney damage induced by exogenous stresses. The aim of this study was to explore whether resveratrol would attenuate concanavalin A (Con A)-induced renal oxidative stress and advanced glomerulosclerosis in aged mice. Aged male C57BL/6 mice were treated orally with resveratrol (30 mg/kg) seven times (12 h intervals) prior to the administration of a single tail-vein injection of Con A (20 mg/kg). The plasma and urinary levels of kidney damage markers were evaluated. The kidney histopathology, renal parameters, and oxidative stress levels were measured. Furthermore, klotho was downregulated in mouse kidney mesangial cells that were pretreated with 25 µM resveratrol followed by 20 µg/mL Con A. The urinary albumin/creatinine ratio, blood urea nitrogen, kidney mesangial matrix expansion, tubulointerstitial fibrosis, and renal levels of α-smooth muscle actin, transforming growth factor beta, fibronectin, procollagen III propeptide, and collagen type I significantly increased in Con A-treated aged mice. Aged mice kidneys also showed markedly increased levels of 8-hydroxydeoxyguanosine (8-OH-dG) and reactive oxygen species (ROS), with reduced superoxide dismutase activity and levels of glutathione, klotho, and SIRT1 after Con A challenge. Furthermore, in kidney mesangial cells, klotho silencing abolished the effects of resveratrol on the Con A-mediated elevation of the indices of oxidative stress and the expression of glomerulosclerosis-related factors. These findings suggest that resveratrol protects against Con A-induced advanced glomerulosclerosis in aged mice, ameliorating renal oxidative stress via the SIRT1-mediated klotho expression.

## 1. Introduction

Aging is a natural progressive event affecting the growth and development of living organisms. However, changes caused by aging represent major risk factors in the development of various diseases of the liver, heart, blood vessels, and kidney [1]. The aged kidney has an increased vulnerability to stress and a decreased ability to recover from acute kidney injury (AKI), further contributing to chronic kidney disease (CKD) [2]. Age-related renal changes include increases in mesangial matrix expansion, glomerular membrane thickening, loss of capillary loops, glomerulosclerosis, interstitial fibrosis, and tubular atrophy [3,4,5]. Therefore, the glomerular filtration rate declines, and the albuminuria and serum creatinine levels commonly increase in older individuals [6]. In addition, aged kidneys show increased levels of oxidative stress, including 8-hydroxydeoxyguanosine (8-OH-dG) and lipid peroxidative damage-associated thiobarbituric acid-reactive substances [7,8]. 

Mesangial cells, the main glomerular cells, participate in the extracellular matrix (ECM) construction in the mesangium and play an important role in regulating glomerular hemodynamics. However, mesangial cells can be activated, resulting in hyperproliferation and excess ECM deposition. Activated mesangial cells can also secrete inflammatory cytokines, adhesion molecules, and chemokines, finally leading to renal glomerular fibrosis [9].

Concanavalin A (Con A) is a lectin utilized for the targeted binding of certain oligosaccharide structures of N-glycosylated proteins [10]. It is well established that macrophage infiltration is involved in concanavalin A (conA)-induced liver injury [11]. Tiegs et al. reported that, compared with control, male NMRI or BALB/c mice developed severe liver injury, with the ALT increasing more than 300-fold after an intravenous dose >1.5 mg/kg concanavalin A (Con A) treatment within 8 h [11]. Con A-induced immune cell-mediated acute hepatitis is a well-established mouse model that mimics human autoimmune hepatitis [11]. Interestingly, our previous experiments had found that Con A challenge not only exacerbated the immune dysregulation and piecemeal necrosis of livers, but also caused renal glomerulosclerosis in aged mice [12]. In addition, Liu et al. also revealed that Con A treatment obviously caused kidney injury in female BALB/C mice through the stimulation of macrophage infiltration [13]. However, whether Con A injection exacerbates kidney damage and induces advanced glomerulosclerosis in aged mice remains unclear. Therefore, in this study, we further investigated whether resveratrol pretreatment protected against the advanced renal glomerulosclerosis in aged mice after Con A challenge.

Sirtuin 1 (SIRT1), a mammalian homolog of yeast silent information regulator 2, is involved in lifespan extension and plays a vital role in aging [14]. It is worth noting that SIRT1 expression is decreased in aging kidneys, and Chuang et al. reported that reduced SIRT1 expression in podocytes aggravates glomerulosclerosis and albuminuria in aged mice and is accompanied by increased urinary 8-OH-dG levels [15]. In addition, nicotinamide mononucleotide, an NAD^+^ precursor, restores age-related sensitivity to AKI through a SIRT1-dependent mechanism by modulating the c-Jun N-terminal kinase signaling pathway [16]. 

Klotho, a transmembrane protein, is an aging suppressor [17] that regulates calcium/phosphorus metabolism [18]. In kidney tissue, klotho is synthesized mainly in the distal renal tubules. A reduced renal klotho expression enhances the high levels of transforming growth factor-β1 (TGF-β1), α-smooth muscle actin (α-SMA), and fibronectin in mice after unilateral ureteral obstruction, which then worsens renal interstitial fibrosis [19]. Lim et al. reported that 24-month-old mice showed lower renal levels of SIRT-1 and klotho than two-month-old mice, as well as greater mesangial volume and tubulointerstitial fibrosis [4]. Furthermore, the reduction of kidney klotho levels might prompt oxidative stress and diminish the expression of superoxide dismutase (SOD), an antioxidant enzyme [4].

Resveratrol, a phenolic compound, can increase the expression of SIRT1 and modulate the inflammatory-oxidative stress cycle by controlling CKD-related metabolic derangements [20]. Moreover, resveratrol enhances creatinine clearance and decreases albuminuria to ameliorate aging-related progressive renal injury in 18-month-old mice via Nrf2-HO1-NOQ1- and SIRT1-AMPK-PGC-1α-mediated signaling pathways [21]. Previous studies indicated that resveratrol enhanced GLUT4 translocation, improved insulin sensitivity, activated SIRT1 and AMPK, decreased adipogenic genes, and reduced diabetic complications [22]. Recently, Espinoza reported that resveratrol binded and inhibited receptor protein kinases (RTK), including Akt, mTOR, JNK, and STAT, which led to apoptosis and inhibited the cell cycle in antitumor mechanisms [23]. As illustrated in our previous study, resveratrol improves Con A-induced hepatitis in aged mice by inducing the SIRT1-mediated repression of p66^shc^ expression [12]. Therefore, in the current study, we investigated whether resveratrol pretreatment protects against Con A-aggravated renal glomerulosclerosis by preventing the reduced expression of SIRT1 and klotho in the same aged mouse model.

## 2. Results

### 2.1. Effects of Resveratrol on Con A-Induced Renal Dysfunctions and Histological Changes in Aged Mice

A histological examination using hematoxylin and eosin staining revealed that the mesangial matrix area was expanded in untreated aged mice compared with that in young mice (Figure 1A). Furthermore, there was extensive renal tubular epithelial cell necrosis, inflammatory cell infiltration, and an increased glomerular basement membrane thickening and mesangial matrix in aged mice challenged with Con A (periodic acid-Schiff stain). These phenomena in Con A-challenged aged mice were attenuated by resveratrol pretreatment. There were no significant differences in the body weights among the four groups, including Yang group, Aged group, Aged + Con A group, and Aged + Con A + RSV group (Figure 1B). Resveratrol did not affect the kidney weight or the kidney weight/body weight ratio in Con A-challenged aged mice (Figure 1C,D).

Aged mice showed a significantly higher urinary albumin/creatinine ratio and BUN than young mice. Both parameters were further significantly increased in aged mice challenged with Con A (Figure 1E,F). Resveratrol administration markedly inhibited the Con A-induced increases in the urinary albumin/creatinine ratio and BUN in aged mice (Figure 1E,F).

### 2.2. Effect of Resveratrol on Con A-Induced Glomerulosclerosis in Aged Mice

Masson-trichrome staining revealed more tubulointerstitial fibrosis and tubular atrophy in the kidneys of aged mice than in those of young mice. Moreover, Con A exacerbated kidney fibrosis and induced vascular sclerosis in aged mice; however, these phenomena were ameliorated by pretreatment with resveratrol (Figure 2A, upper panel). In parallel, after the images of α-SMA IHC staining were quantified in the kidney tissues of mice, the results showed that the renal protein level of α-SMA was significantly reduced in Con A-challenged aged mice after resveratrol pretreatment (Figure 2A, middle and lower panels). In addition, after Con A stimulation, there were increases in the renal mRNA levels of TGF-β, fibronectin, procollagen-III, and collagen I, four glomerulosclerosis-related factors, in aged mice. These increases were suppressed by resveratrol administration (Figure 2B).

### 2.3. Effect of Resveratrol on Klotho and SIRT1 in Con A-Challenged Aged Mice

The results of IHC staining showed that the renal levels of klotho and SIRT1 were significantly lower in aged mice than in young mice, and a further reduction in klotho and SIRT1 expression occurred in Con A-challenged aged mice (Figure 3A). Conversely, the administration of resveratrol prevented the Con A-induced reduction of klotho and SIRT1 levels in the kidney tissues of aged mice (Figure 3A). Similar changes were observed in the mRNA levels of klotho and SIRT1 in the kidney tissues of mice (Figure 3B).

### 2.4. Effect of Resveratrol on Con A-Induced Oxidative Stress in the Kidney Tissues of Aged Mice

IHC staining revealed that the renal 8-OH-dG levels were higher in aged mice than in young mice (Figure 4A). This increase was enhanced in aged mice challenged with Con A. Resveratrol pretreatment significantly inhibited the Con A-induced increase of 8-OH-dG in kidneys from aged mice (Figure 4A). A similar increase was observed in the ROS levels in the kidney tissues of aged mice injected with Con A (Figure 4B). In addition, the MDA level was significantly higher in aged than in young mice, and a further elevation occurred in Con A-challenged aged mice (Figure 4C). Con A challenge also enhanced the aged-related reduction of the SOD activity and GSH levels in kidney tissues (Figure 4D,E). Importantly, these phenomena in Con A-treated mice were abolished by resveratrol pretreatment (Figure 4).

### 2.5. Effects of Resveratrol and Klotho Gene Silencing on the Con A-Mediated Reduction of Klotho and SIRT1 Levels and the Increased Expression of Glomerulosclerosis-Related Factors and ROS in Mesangial Cells

To explore whether the reduction of the renal levels of klotho and SIRT1 directly increased glomerulosclerosis and oxidative stress in Con A-challenged aged mice, and whether this effect would be blocked by pretreatment with resveratrol, mouse kidney mesangial cells were cultured in a Con A-containing medium. Indeed, resveratrol pretreatment (25 µM) prevented Con A-induced increases in the mRNA levels of TGF-β, fibronectin, procollagen-III, and collagen I (Figure 5A), and ROS levels (Figure 5B) were also significantly diminished by pretreatment with 25 μM resveratrol.

The mesangial cells were transient transfected with klotho siRNA for 60 h. Subsequently, the mRNA levels of TGF-β, fibronectin, procollagen-III, and collagen I (Figure 5C), and the relative ROS levels were measured using flow cytometry (Figure 5D) after serial treatments with klotho siRNA, 25 μΜ resveratrol for 2 h, and 25 μΜ resveratrol plus 20 μg/mL Con A for 30 min. The results are expressed as means ± S.E.M. from at least three independent experiments. Different letters between groups indicate statistically significant differences (*p* < 0.05).

Subsequently, we decreased the klotho levels using siRNA to confirm a direct relationship between klotho, glomerulosclerosis, and oxidative stress and Con A challenge. The protein level of klotho was reduced by approximately 70% in mesangial cells incubated with 100 nM klotho siRNA (the data were not shown). Preincubation with klotho siRNA blocked the ability of 25 μM resveratrol to inhibit the increased mRNA levels of TGF-β, fibronectin, procollagen, and collagen 1, as well as ROS levels, caused by 20 μg/mL Con A (Figure 5C,D).

## 3. Discussion

Aged kidneys are characterized by glomerulosclerosis, interstitial fibrosis, tubular atrophy, and loss of kidney function, leading to an age-related increase in the incidence of CKD [24]. The kidneys exhibit a greater age-associated deterioration than other organs. This renal aging results in reduced capacities for adaption to stress and structural repair [25]. Phenomena of age-related kidney injury include the enlargement of the mesangial compartment in the glomerulus, development of glomerulosclerosis in an increasing proportion of glomeruli, and reductions in the glomerular filtration rate and capacity to concentrate urine [26]. Therefore, understanding how aging affects the glomerulus will improve our ability to recognize the risk for kidney damage and progression to AKI or CKD in the older population. 

Resveratrol has a protective effect in controlling CKD-related metabolic derangements. The present study verified that resveratrol administration attenuated Con A-exacerbated albuminuria, oxidative stress (increased levels of 8-OH-dG, MDA, and ROS, and a decreased activity of SOD and GSH), and glomerulosclerosis (Masson’s trichrome stain and increased levels in α-SMA, TGF-β, fibronectin, procollagen-III, and collagen I) in the kidney tissues of aged mice, possibly by preventing reductions in klotho and SIRT1 levels. The roles of klotho and oxidative stress in glomerulosclerosis were further supported by in vitro studies using mesangial cells treated with resveratrol and/or via the silencing of klotho prior to Con A challenge. Few studies have explored whether Con A exacerbates kidney damage and induces advanced glomerulosclerosis with aging. To the best of our knowledge, this is the first study to show that resveratrol pretreatment prevents Con A-aggravated advanced glomerulosclerosis in aged mice.

The injection of the T cell mitogen, Con A, into mice induces an acute, T cell-dependent autoimmune hepatitis [11]. The aging process is considered a high-risk factor for chronic diseases, especially autoimmune disorders. Our previous report revealed that a dramatic elevation of the immune response and inflammation in plasma and liver tissue occurred in aged mice treated with Con A, which exacerbated the progression of hepatitis [12]. Importantly, pretreatment with resveratrol reversed these phenomena [12]. 

Although Liu et al. reported that mouse kidneys were damaged (characterized by extensive renal tubular epithelial cell necrosis and proinflammatory cytokine infiltration) after Con A administration [13], whether Con A injection exacerbated age-mediated kidney injury and severe nephropathy in aged rodents remained unclear. The present results show that, compared to those of young mice (four to six months of age), the kidneys of aged mice (24 months of age) had an expanded mesangial matrix area and some tubulointerstitial fibrosis accompanied by partial inflammation and tubular atrophy. Furthermore, there were significant enhancements of glomerular basement membrane thickening, renal fibrosis, vascular sclerosis, and inflammatory cell infiltration in the kidneys of Con A-challenged aged mice (Figure 1A and Figure 2A); this phenomenon is known as glomerulosclerosis.

Albuminuria reflects damage to the glomerular filtration barrier and is accompanied by glomerulosclerosis [27]. Liu et al. suggested that the BUN and serum creatinine levels were significantly increased in BALB/c mice 8 h after Con A administration [13]. Correspondingly, Con A-treated aged mice also showed higher urinary albumin/creatinine ratios and BUN levels than control aged mice (Figure 1E,F). A previous study showed that renal fibrosis caused by a unilateral ureteral obstruction in mice resulted in increased levels of TGF-β1, α-SMA, and fibronectin, as well as a reduced level of klotho [19]. Tissue fibrosis is characterized by an excessive synthesis and deposition of ECM through the induction of procollagen-III and collagen I [28]. Likewise, we found that the renal levels of procollagen-III and collagen I were markedly enhanced in aged mice treated with Con A. Together, these results demonstrate that aging increased the susceptibility to advanced glomerulosclerosis in the kidneys of mice challenged with Con A. Importantly, we observed that these phenomena were attenuated by resveratrol pretreatment.

Excessive oxidative stress is critically involved in the overall aging process. The decline in renal function that occurs in a large proportion of the aging and CKD populations may be linked to increasing levels of oxidative stress and inflammation. Glomerulosclerosis is commonly observed in adults with diabetic nephropathy. Wu et al. reported that severe albuminuria, an elevated serum creatinine level, and glomerulosclerosis developed in 18-month old mice after the induction of diabetes with streptozotocin, which were phenomena not observed in young or untreated aged mice; furthermore, they observed that these symptoms were associated with elevated oxidative stress [29]. Indeed, our results show that a dramatic elevation in the renal levels of 8-OH-dG, MDA, and ROS occurred in the kidneys of aged mice challenged with Con A and was accompanied by a decrease in the enzymatic activity of SOD and GSH levels (Figure 4).

ROS, which are formed in cells by various environmental agents and the endogenous oxygen metabolism, can damage DNA. 8-OH-dG is a major DNA damage product and is considered a marker of cellular oxidative stress [30]. Partial knockdown of SIRT1 in renal podocytes intensifies glomerulosclerosis in aged mice and is accompanied by an increased urinary 8-OH-dG level [15]. Therefore, we considered 8-OH-dG to be a significant indicator of aging in kidneys and an increased prevalence of specific kidney diseases. In our study, resveratrol administration diminished Con A-induced renal oxidative stress through the upregulation of antioxidants, including SOD and GSH, and ameliorated glomerulosclerosis and kidney damage in aged mice. Similar results were reported by Jang et al., who found that resveratrol treatment attenuated albuminuria and renal tubulointerstitial fibrosis in aged mice through a reduction of the increase in the renal levels of 8-OH-dG, 3-nitrotyrosine, collagen IV, and fibronectin [31]. 

Klotho is an aging suppressor gene. A previous study showed that increasing the klotho protein level inhibited oxidative stress at the cellular level in mammals [32]. Wu et al. found that klotho reduced fibrosis and the expression of inflammatory cytokines in a high-glucose-treated rat mesangial cell model via the suppression of the Egr1/TLR4/mTOR axis [33]. In addition, resveratrol increased the mRNA and protein levels of klotho in mouse kidneys, in both in vivo and in vitro models, by stimulating the SIRT1 upstream pathway [34]. In this study, our IHC staining results also found that the renal protein levels of klotho and SIRT1 were markedly decreased in aged mice treated with Con A, but these changes were reversed by resveratrol pretreatment (Figure 3). Therefore, we speculate that Con A-induced severe albuminuria and advanced glomerulosclerosis are highly correlated with increased renal oxidative stress via the repression of SIRT1-mediated klotho expression and that these effects are attenuated by resveratrol pretreatment.

Currently, many rodent renal failure models have been established to advance the understanding of human nephropathy, such as 5/6 nephrectomy, adenine diet feeding, and unilateral ureteral obstruction (UUO). The 5/6 subtotal nephrectomy approach, a remnant kidney model, is used more in rodents to mimic human chronic kidney disease. After a nephrectomy in rodents, glomerular hypertension was induced by the stimulation of the renin angiotensin system [35], and furthermore it caused glomerulosclerosis, renal atrophy, and proteinuria [36]. In addition, a rat model who was fed a 0.75% adenine diet for four weeks was reported on in order to study kidney damage that paralleled chronic kidney disease in humans [37], because adenine enhanced the increase of the BUN and serum creatinine level, as well as inducing kidney atrophy and fibrosis [38]. UUO is an experimental model of renal injury that is characterized by progressive tubulointerstitial fibrosis and that resembles clinical obstructive nephropathy. It was shown that UUO challenge induced the mechanisms of apoptosis, inflammation, and fibrosis in the kidney tissues of a rodent model and further resulted in renal hemodynamic impairment [39]. Although the studies on Con A-induced renal glomerulosclerosis in rodent models were rare, we speculated that Con A-exacerbated renal glomerulosclerosis in aged mice may be related to immune dysregulation, according to Liu et al.’s study [13]. In order to be proven, this concept will require extensive experimentation in the future.

A limitation of our study is the use of resveratrol pretreatment. Thus, further work is required to validate the ability of resveratrol to reverse existing Con A-aggravated glomerulosclerosis in aged kidneys. In addition, studies using SIRT1- and klotho-deficient knockout mice are needed to verify a direct or indirect effect of resveratrol. To partially address these limitations, we used mouse kidney mesangial cells to demonstrate the mechanism of the inhibitory effect of resveratrol pretreatment against Con A-induced increases of ROS and glomerulosclerosis-related factor expression through the downregulation of the klotho level (Figure 5).

## 4. Materials and Methods

### 4.1. Animals

The animal study was performed following our previous procedures [12], with slight modifications. A total of 32 male C57BL/6 mice were purchased from the National Laboratory Animal Center (Taipei, Taiwan), and the mice were divided into four groups: (1) 4- to 6-month-old untreated (“young”) mice (*n* = 8), (2) 24-month-old untreated (“aged”) mice (*n* = 8), (3) Con A-challenged aged mice (*n* = 8), and (4) resveratrol-pretreated + Con A-challenged aged mice (*n* = 8). Mice were administered seven doses of the 0.5% (*w*/*v*) sodium carboxymethyl cellulose vehicle or 30 mg/kg body weight of resveratrol (Sigma-Aldrich, St. Louis, MO, USA) orally at 12 h intervals. Twelve hours after the last oral resveratrol treatment, the group 3 and 4 mice were treated with 20 mg/kg Con A (Sigma-Aldrich, St. Louis, MO, USA) by injection into the tail vein. Untreated young and aged mice were given pyrogen-free saline at an equal volume. Each mouse was placed in an individual metabolic cage, and 12-h urine was collected. Afterwards, the mice were anesthetized by intramuscular injection with 15 mg/kg Zoletil (Virbac, Carros, France) before decapitation to obtain blood and kidney tissues for further analysis. All protocols were performed according to the Guide for the Care and Use of Laboratory Animals and were approved by the Institutional Animal Care and Use Committee (IACUC) of China Medical University (Protocol 104-34-C-1, permission code, December 2014).

### 4.2. Mouse Glomerular Mesangial Cell Culture

The mouse glomerular mesangial cell line was obtained from the Bioresource Collection and Research Center, Food Industry Research and Development Institute (cat. No. 60366; Hsinchu, Taiwan). Cells were cultured in a 3:1 mixture of Dulbecco’s modified Eagle’s medium and Ham’s F12 supplemented with 5% fetal bovine serum, 14 mM HEPES, and 1% penicillin/streptomycin in a humidified 5% CO_2_ incubator at 37 °C.

### 4.3. Measurements of Blood Urea Nitrogen (BUN) and Urinary Concentrations of Albumin and Creatinine

Urinary creatinine levels were measured by a colorimetric assay kit (Cayman Chemical, Ann Arbor, MI, USA). Urinary albumin concentrations were detected using a mouse albumin enzyme-linked immunosorbent assay kit (Abcam, Cambridge, UK). The urinary albumin/creatinine ratio was calculated as: albumin (μg/mL)/creatinine (μg/mL) [40]. The BUN concentration was detected using a colorimetric kit (Thermo Fisher Scientific, Waltham, MA, USA). All protocols were carried out according to the manufacturers’ instructions.

### 4.4. Histological Staining

Kidney histological assays were performed with hematoxylin and eosin, immunohistochemical (IHC), immunofluorescence, and Masson’s trichrome stains, according to our previous protocols [12,41]. Briefly, kidney tissues were soaked in 4% paraformaldehyde, embedded in paraffin, and then cut into 5-μm-thick sections. The sections were stained with hematoxylin, eosin, and Masson’s trichrome (Sigma-Aldrich) for histopathology analyses. In addition, the sections were deparaffinized and incubated in 0.3% H_2_O_2_ with primary antibodies, including anti-klotho, anti-SIRT-1, anti-α-SMA (all from Abcam, Cambridge, UK), and anti-8-OHdG (Santa Cruz Biotechnology, Dallas, TX, USA). Subsequently, biotinylated secondary antibodies and the avidin-biotin complex reagent were added, and a color development was achieved with 3,3′-diaminobenzidine for the IHC stain. For the reactive oxygen species (ROS) immunofluorescence (IF) staining, 2′,7′-dichlorofluorescein diacetate (Thermo Fisher Scientific) was used, and the fluorescent oxidized product was visualized using an Olympus U-RFL-T Fluorescence Microscope System (Olympus Life Science, Tokyo, Japan). Slides (5 µm) were also stained with periodic acid-Schiff (Sigma-Aldrich) to observe the thickening of the glomerular and tubular basement membranes, as well as the amount of mesangial matrix. The images were visualized and quantified using the Panthera L Smart Light Microscope system (Motic, San Antonio, TX, USA).

### 4.5. Reverse Transcription Quantitative Polymerase Chain Reaction

Total RNA was extracted from kidney tissues and glomerular mesangial cells, and a reverse transcription quantitative polymerase chain reaction was performed according to our previous protocol [12]. Primer sequences are listed in Table 1.

### 4.6. Lipid Peroxidation Assay

A renal lipid peroxidation assay was carried out, and the results are expressed as malondialdehyde (MDA) equivalents (nmol/mg protein), according to our previous protocol [12]. 

### 4.7. Measurements of SOD Activity and Glutathione (GSH) Levels 

The renal activity of SOD and renal GSH levels were detected using SOD and GSH assay kits (Cayman Chemical, Ann Arbor, MI, USA), according to our previous protocols [12].

### 4.8. Flow Cytometry for ROS 

Glomerular mesangial cells were treated with 10 μM 2′,7′-dichlorofluorescein diacetate (Thermo Fisher Scientific, Waltham, MA, USA) for 40 min. Upon stopping the reaction, the cells were washed gently with ice-cold phosphate-buffered saline. Dichlorofluorescein fluorescence was detected by flow cytometry using a FACSCalibur system (BD Biosciences, Franklin Lakes, NJ, USA). Data are presented as relative fold changes in fluorescence.

### 4.9. Klotho Small Interfering (Si) RNA Transient Transfection

Klotho siRNA transient transfection was performed according to our previous protocol [12]. The specific siRNA duplex nucleotide for klotho was designed as 5′-UGCGCAAGGUCUCCGGUACUA-3′. The scramble control siRNA duplex nucleotide was 5′-UGCGCGAAGUCUCCGGUACUA-3′.

### 4.10. Statistical Analyses

Data are presented as means ± S.E.M. *p* < 0.05 denotes a statistically significant difference. Comparisons among groups were performed using a one-way analysis of variance and subsequent Tukey’s post-hoc test for multiple comparisons.

## 5. Conclusions

Our data indicated that resveratrol pretreatment protected against Con A-induced advanced glomerulosclerosis in aged mice by attenuating renal oxidative stress, at least in part, through SIRT1-mediated klotho expression. Meanwhile, a schematic diagram of this study is shown in Figure 6. Therefore, we suggest that resveratrol can be used as a therapeutic strategy for aging-mediated CKD, although this possibility requires additional confirmation through further research and further efficacy evaluations.

## Figures and Tables

**Figure 1 ijms-21-06766-f001:**
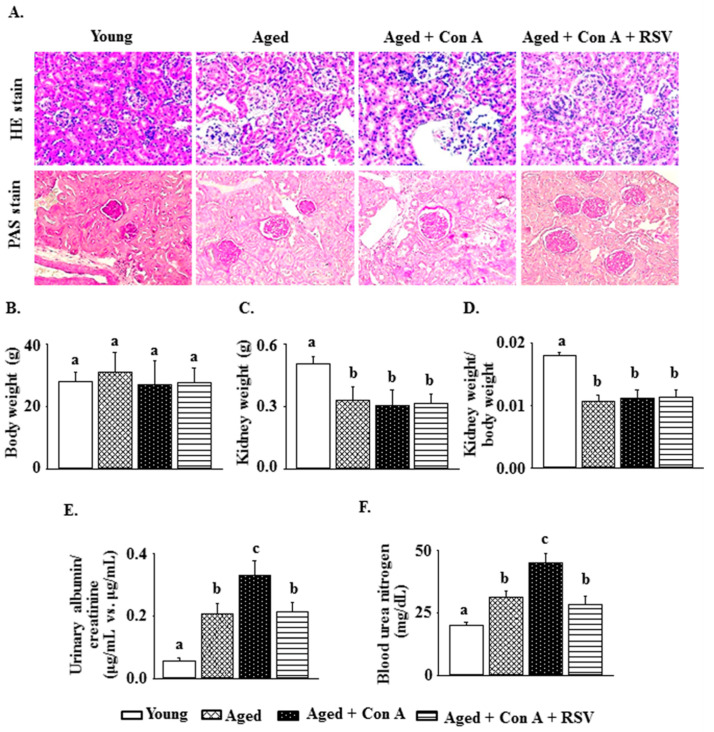
Resveratrol attenuates concanavalin A (Con A)-aggravated renal damage in aged mice. (**A**) Mouse kidney tissues were stained with hematoxylin and eosin (upper panel, 200×) and periodic acid-Schiff (lower panel, 200×). (**B**) Body weights, (**C**) kidney weights, and (**D**) kidney weight/body weight ratios. (**E**) Albumin/creatinine ratios in urine after 12-h urine collection. (**F**) Blood urea nitrogen (BUN) was detected using a colorimetric BUN determination assay kit. Values are expressed as means ± S.E.M. (each group contains 7–8 mice). Different letters (a, b, and c) indicate significant differences between the groups (*p* < 0.05).

**Figure 2 ijms-21-06766-f002:**
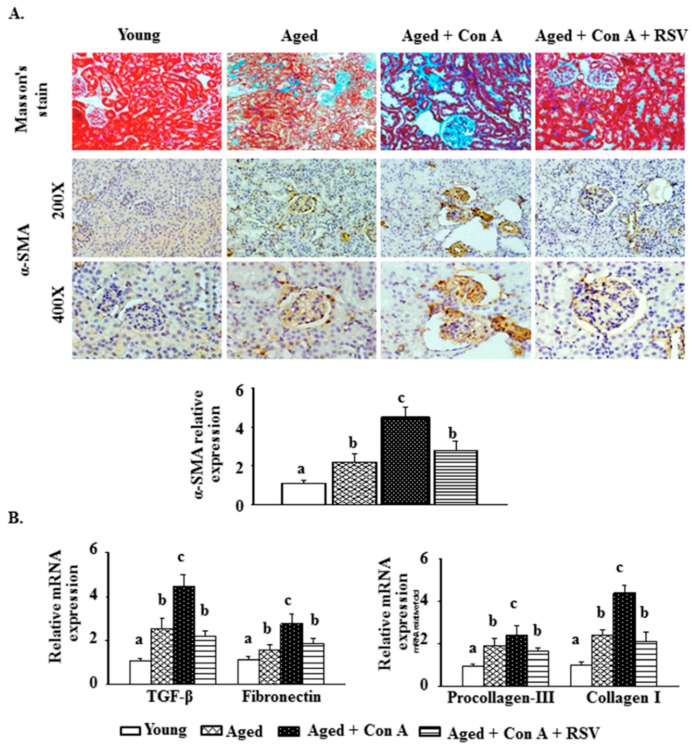
Resveratrol ameliorates concanavalin A (Con A)-induced glomerulosclerosis in the kidney tissues of aged mice. (**A**) Mouse kidney tissues were stained with Masson’s trichrome (upper panel, 100×) and an α-SMA antibody (middle and lower panels, magnification at 200× and 400×, respectively). The images of α-SMA IHC staining at 200× magnification were quantified using the Panthera L Smart Light Microscope system. (**B**) Renal mRNA levels of transforming growth factor (TGF)-β, fibronectin, procollagen-III, and collagen I. Results are expressed as means ± S.E.M. (each group contains 7–8 mice). Different letters (a, b, and c) indicate significant differences between the groups (*p* < 0.05).

**Figure 3 ijms-21-06766-f003:**
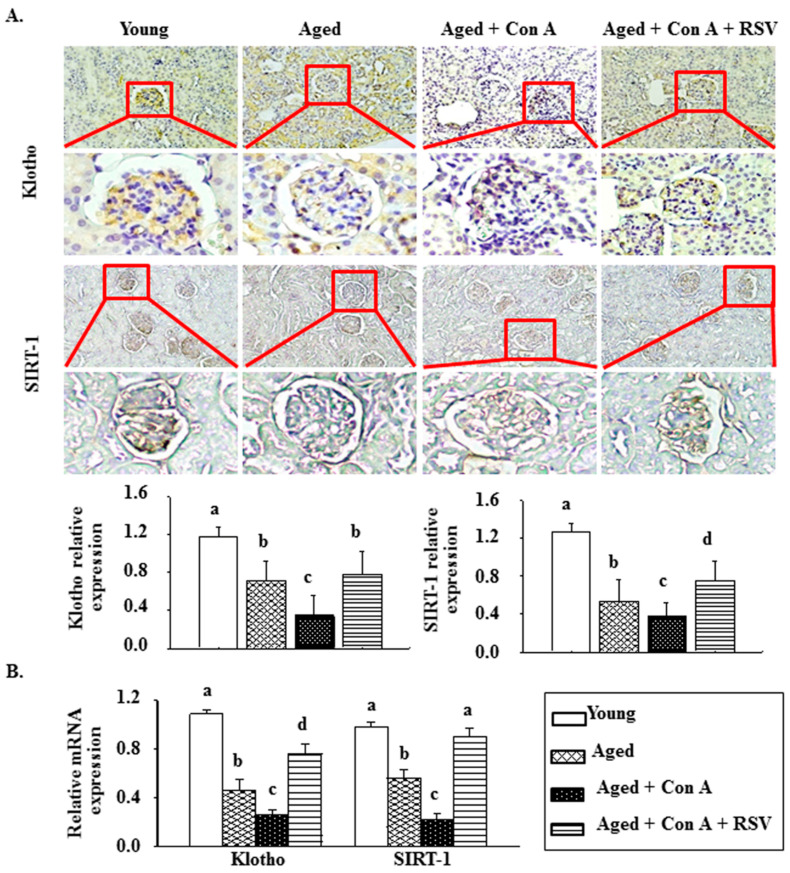
Effects of resveratrol on the renal levels of klotho and SIRT-1 in aged mice challenged with concanavalin A (Con A). (**A**) Representative immunohistochemistry results for klotho and SIRT-1. The images were shown at a 200× magnification, with the lower panels at a 400x enlargement for selected squares, respectively. The results of IHC staining were quantified using the Panthera L Smart Light Microscope system. (**B**) The mRNA levels of klotho and SIRT-1 in mouse kidney tissues were measured by quantitative real-time polymerase chain reaction and western blot, respectively. Data are shown as means ± S.E.M. (each group contains 7–8 mice). Different letters (a, b, c, and d) indicate significant differences between the groups (*p* < 0.05).

**Figure 4 ijms-21-06766-f004:**
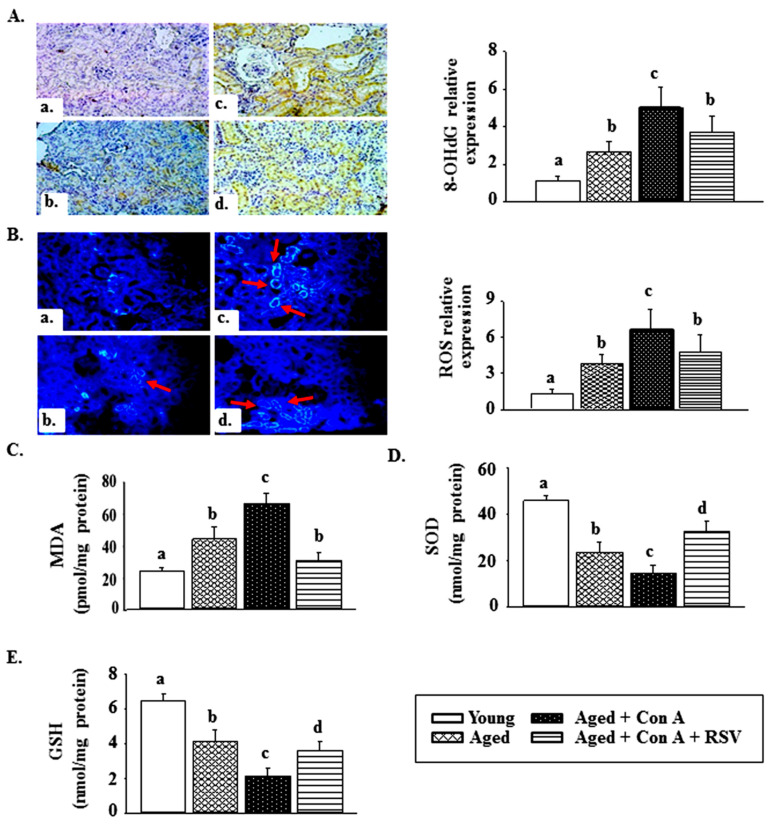
Effects of resveratrol on oxidative stress in kidney tissues from aged mice treated with concanavalin A (Con A). (**A**) Mouse kidney tissues were stained with an 8-hydroxydeoxyguanosine (8-OHdG) antibody (200×). a: young mice, b: aged mice, c: aged mice treated with Con A, d: aged mice treated with Con A + resveratrol (RSV). (**B**) The reactive oxygen species (ROS) level is shown as dichlorofluorescein fluorescence. Red arrows indicate the sites of ROS. a: young mice, b: aged mice, c: aged mice treated with Con A, d: aged mice treated with Con A + RSV. The results of 8-OHdG IHC staining and ROS IF staining were quantified using the Panthera L Smart Light Microscope system. (**C**) Renal lipid peroxidation is shown as malondialdehyde (MDA) equivalents. (**D**) Superoxide dismutase (SOD) activities and (**E**) glutathione (GSH) levels in kidney tissues were measured using kinetic colorimetric assay kits. Results are expressed as means ± S.E.M. (each group contains 7–8 mice). Different letters (a, b, c, and d) indicate significant differences between the groups (*p* < 0.05).

**Figure 5 ijms-21-06766-f005:**
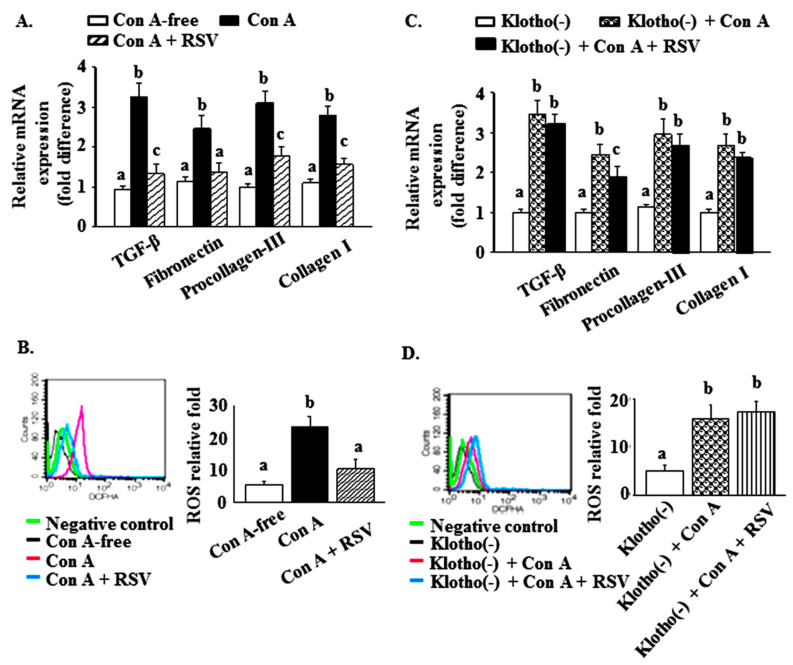
Effects of resveratrol and klotho gene silencing on concanavalin A (Con A)-induced changes in the protein levels of klotho and SIRT-1, glomerulosclerosis-regulated factors, and reactive oxygen species (ROS) in mesangial cells. (**A**) The mRNA levels of transforming growth factor (TGF)-β, fibronectin, procollagen-III, and collagen I, and (**B**) the relative ROS levels were detected using flow cytometry after preincubation of mesangial cells with 25 μM resveratrol for 2 h, followed by cotreatment with 25 μΜ resveratrol plus 20 μg/mL Con A for 30 min. (**C**) mRNA levels of TGF-β, fibronectin, procollagen-III and collagen I, and (**D**) ROS relative levels were measured using flow cytometry after sequential treatments with klotho siRNA, 25 μΜ resveratrol for 2 h, 25 μΜ resveratrol plus 20 μg/ml Con A for 30 min. Different letters (a, b, and c) indicate significant differences between the groups (*p* < 0.05).

**Figure 6 ijms-21-06766-f006:**
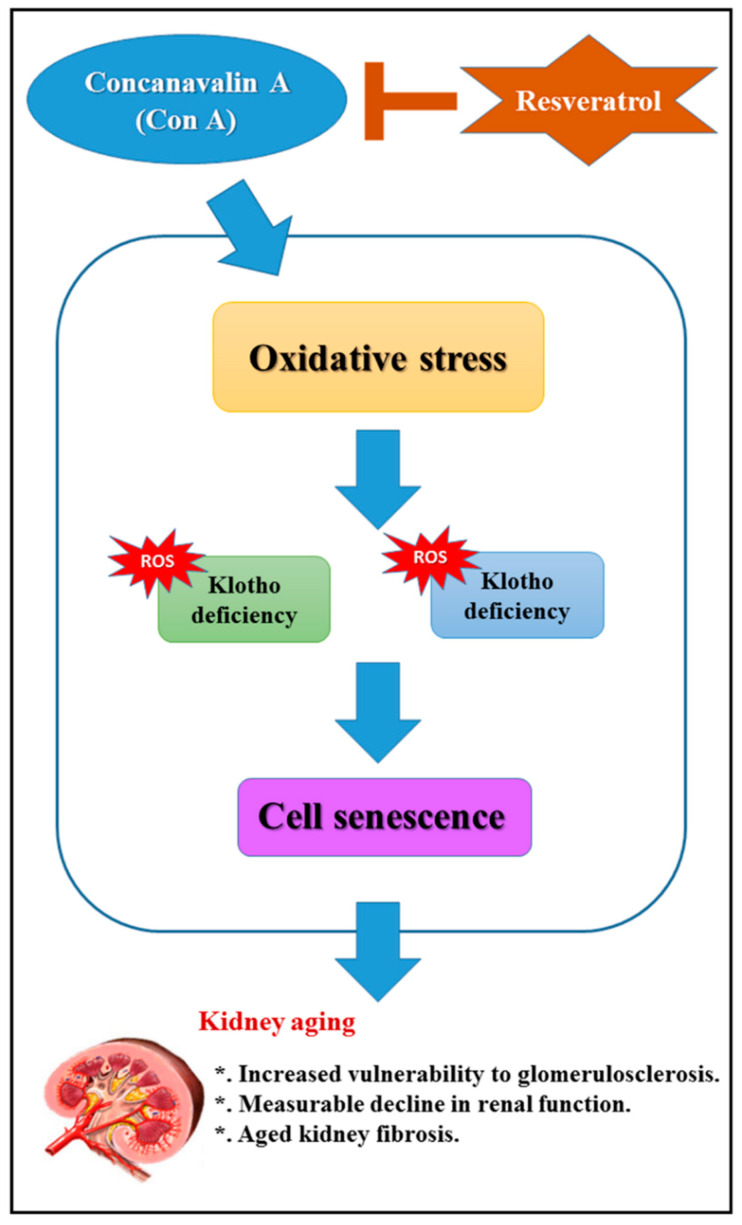
Schematic diagram of resveratrol pretreatment to prevent against concanavalin A (Con A)-induced advanced renal oxidative stress and glomerulosclerosis in aged mice.

**Table 1 ijms-21-06766-t001:** Primer sequences used for reverse transcription quantitative PCR.

Gene	Forward	Reverse
Klotho	AGACCTCCCGATGTATGTGAC	CGAGATGAAGACCAGCAAAG
SIRT-1	GCAACAGCATCTTGCCTGA	GTGCTACTGGTCTCACTT
TGF-β	TGCCCTCTACAACCAACACAACCCG	AACTGCTCCACCTTGGGCTTGCGAC
Fibronectin	TAGGAGAACAGTGGCAGAAAG	CCATCGGGACTGGGTTCA
Procollagen-III	CCCCTGGTCCCTGCTGTGG	GAGGCCCGGCTGGAAAGAA
Collagen I	GAGAGGTGAACAAGGTCCCG	AAACCTCTCTCGCCTCTTGC
GAPDH	TCACCACCATGGAGAAGGC	GCTAAGCAGTTGGTGGTGCA

SIRT-1, sirtuin 1; TGF-β, transforming growth factor-β; GAPDH, glyceraldehyde 3-phosphate dehydrogenase.

## Data Availability

The datasets used and/or analyzed during the current study are available from the corresponding author upon reasonable request.

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
