# Peer review of "Resveratrol Pretreatment Ameliorates Concanavalin A-Induced Advanced Renal Glomerulosclerosis in Aged Mice through Upregulation of Sirtuin 1-Mediated Klotho Expression"

_ijms, 2020, doi:10.3390/ijms21186766_

Round 1

Reviewer 1 Report

this original research paper is interesting and well written

in legend figures please indicate the statistical significance of letters a b c d in each figure

please justify age and sex of mice used in the study

in discussion please discuss papers describing other mirnas in the same  category of disease 

please add a final recapitulate figure

Author Response

Reviewer # 1 Comments:

  1. This original research paper is interesting and well written.

Response:

We are very grateful for the support from the reviewer and thank you very much for everything.

  1. In legend figures please indicate the statistical significance of letters a b c d in each figure.

Response:

We thank the reviewer for drawing our attention to this point. Actually, we had wrote a sentence to describe it in the end of figure legend, “Different letters between groups indicate statistically significant differences (p < 0.05)”. I would like to mark significant differences between two groups with different letters (like Figure 1-E, group Young: “a” and group Aged: “b”). If there is no significant differences between two groups they get the same letter (like Figure 1-E, group Aged: “b” and group Aged + Con A + RSV: “b”).

To makes it clearly, we revised the sentence from “Different letters between groups indicate statistically significant differences (p < 0.05)” to “Different letter indicate significant differences between the groups (p < 0.05)”. (Page 4, line 118; page 5, line 141; page 6, line 156; page 7, line 179; page 8, line 194 with blue color)

  1. Please justify age and sex of mice used in the study.

Response:

We do appreciate for the helpful comments. Actually, we had wrote a sentence to describe it at the part of Animal in the section of Materials and Methods (page 11, line 325 to line 328). We also added some detail information in the section of Materials and Methods (page 11, line 325 with blue color).

  1. In discussion please discuss papers describing other miRNAs in the same category of disease.

Response:

Thanks for the Reviewer's suggestion. Maybe we have misunderstanding this meaning of the Reviewer's comment, because our study has not mention any information about the role of miRNAs in Con A-induced advanced renal glomerulosclerosis in aged mouse model. Could the Reviewer provide more information for us, we will definitely response to this comment in detail as soon as possible.

  1. Please add a final recapitulate figure.

Response:

We do appreciate for the helpful comments. We had revised the section of Conclusions (page 10, line 313, 315 and page 11, line 320 to 321 with blue color) and took the figure 6 as the final recapitulate figure.

Reviewer 2 Report

The authors studied the effect of concanavalin-A (Con-A) on kidney function in aged mice, the  protective effect of resveratrol, and the role of KLOTHO. 

  1. Please provide more information on Con-A as an established mouse model of kidney disease. Add references in the introduction, and compare this mouse model to more conventional mouse models of CKD (e.g. 5/6th nephrectomy, adenine, ureter ligation, ..). The authors do not provide a clear reason for selecting this model
  2. Please add values of creatinine as well. What is the evolution over time, after injection of Con-A? is this a model of transient increases in a number of markers, or does it lead to longer-term alterations in kidney function. 
  3. Con-A is a model of acute hepatitis. PLease add data on this to the manuscript.
  4. Was this a separate study, or a re-analysis of left-over tissue from other studies. 
  5. The rationale for resveratrol needs to be explained in greater detail. 

Author Response

Reviewer # 2 Comments:

The authors studied the effect of concanavalin-A (Con-A) on kidney function in aged mice, the protective effect of resveratrol, and the role of KLOTHO.

  1. Please provide more information on Con-A as an established mouse model of kidney disease. Add references in the introduction, and compare this mouse model to more conventional mouse models of CKD (e.g. 5/6th nephrectomy, adenine, ureter ligation, ..). The authors do not provide a clear reason for selecting this model.

Response:

We would like to thank the reviewer for the comment. We had some description in the section of Introduction (page 2, line 61 to line 73 with blue color) and the section of discussion (page 9, line 289 to 304 with blue color). And we also added some references [35] to [39] in the manuscript (page 15, line 506 to 517 with blue color). We revised the order of references.

[36]. Gong, W.; Mao, S.; Yu, J.; Song, J.; Jia, Z.; Huang, S.M.; Zhang, A.H. NLRP3 deletion protects against renal fibrosis and attenuates mitochondrial abnormality in mouse with 5/6 nephrectomy. Am J Physiol Renal Physiol. 2016, 310, F1081-1088.

[37]. Yokozawa, T.; Zheng, P.D.; Oura, H.; Koizumi, F. Animal model of adenine-induced chronic renal failure in rats. Nephron 1986, 44, 230-234.

[38]. López-Novoa, J.M.; Rodríguez-Peña, A.B.; Ortiz, A.; Martínez-Salgado, C.; López Hernández, F.J. Etiopathology of chronic tubular, glomerular and renovascular nephropathies: clinical implications. J Transl Med. 2011, 9, 13.

[39]. Chevalier, R.L.; Forbes, M.S.; Thornhill, B.A. Ureteral obstruction as a model of renal interstitial fibrosis and obstructive nephropathy. Kidney Int. 2009, 75, 1145-1152.

  1. Please add values of creatinine as well. What is the evolution over time, after injection of Con-A? is this a model of transient increases in a number of markers, or does it lead to longer-term alterations in kidney function.

Response:

We would like to thank the reviewer for the comment. First, as shown as below, that is the creatinine level in our mouse model. Our data indicated that the urinary creatinine level decreased by Con A in aged mice. We already showed the ratio of urinary albumin/creatinine in the manuscript to demonstrate the kidney function in our animal model. If editor think that is necessary, we will add the creatinine data in the manuscript.

Second, we had some description in the section of discussion (page 10, line 301 to 304 with blue color). Based on our best knowledge, Con-A-induced renal glomerulosclerosis in rodent model are rare. So far, it is not easy to draw a conclusion that Con-A leads to longer-term alterations in kidney function or not. This concept is required our extensive experiment to proof in the future.

  1. Con-A is a model of acute hepatitis. Please add data on this to the manuscript.

Response:

We would like to thank the reviewer for the comment. We had some description in the section of Introduction (page 2, line 61 to 64 with blue color).

  1. Was this a separate study, or a re-analysis of left-over tissue from other studies.

Response:

We would like to thank the reviewer for the comment. This is a separate study completed in China Medical University and Chang Gung Memorial Hospital.

  1. The rationale for resveratrol needs to be explained in greater detail.

Response:

We would like to thank the reviewer for the comment. We had added some description in the section of Introduction (page 2, line 93 to page 3, line 97 with blue color). And we also added two references [22] and [23] in the manuscript (page 15, line 475 to 478 with blue color). We revised the order of references.

[22]. Bagul PK, Banerjee SK. Application of resveratrol in diabetes: rationale, strategies and challenges. Curr Mol Med. 2015;15(4):312-30.

[23]. Espinoza JL, Kurokawa Y, Takami A. Rationale for assessing the therapeutic potential of resveratrol in hematological malignancies. Blood Rev. 2019 Jan;33:43-52.

Round 2

Reviewer 1 Report

changes are ok